# The Role of Dynamic Computed Tomography Angiography in Endoleak Detection and Classification After Endovascular Aneurysm Repair: A Comprehensive Review

**DOI:** 10.3390/diagnostics15030370

**Published:** 2025-02-04

**Authors:** Alexandra Catasta, Claudio Bianchini Massoni, Davide Esposito, Sara Seitun, Giovanni Pratesi, Nicola Cicala, Antonio Freyrie, Paolo Perini

**Affiliations:** 1Vascular Surgery, Cardio-Thoracic and Vascular Department, University-Hospital of Parma, 43126 Parma, Italy; claudiobianchinim@gmail.com (C.B.M.); nicola.cicala21@gmail.com (N.C.); antonio.freyrie@unipr.it (A.F.); 2Department of Surgical and Integrated Diagnostic Sciences, University of Genoa, 16132 Genoa, Italy; davide.esposito11@gmail.com (D.E.); giovanni.pratesi@unige.it (G.P.); 3Clinic of Vascular and Endovascular Surgery, IRCCS Ospedale Policlinico San Martino, 16132 Genoa, Italy; 4Department of Radiology, IRCCS Ospedale Policlinico San Martino, 16132 Genova, Italy; saraseitun@yahoo.com; 5Department of Medicine and Surgery, University of Parma, 43126 Parma, Italy

**Keywords:** endoleak, EVAR, four-dimensional computed tomography, 4D, dynamic computed tomography

## Abstract

**Backgroud:** The use of dynamic computed tomography angiography (dCTA) for the detection of endoleaks in patients who underwent endovascular repair of abdominal aortic aneurysms is gaining interest. This study aims to provide an overview of the current applications of dCTA technologies in vascular surgery. **Methods**: We performed a comprehensive review by searching in the PubMed database and Cochrane Library (last search: 1 November 2024). We included studies considering endoleak investigation after endovascular aneurysm repair (EVAR). We included papers that reported the outcome of applications of dCTA, excluding case reports or very limited case series (≤4). Finally, 14 studies regarding 377 computed tomography angiographies (CTA) were included and evaluated. **Results**: Persistent perfusion of the aneurysm sac is the most common complication after EVAR. Imaging-based surveillance post-EVAR is essential with the aim of early detection, characterization, and localization of endoleaks to guide therapeutic intervention or follow-up. dCTA detected 36 type I endoleaks versus 16 identified with standard CTA and 138 versus 95 type II endoleaks. **Conclusions:** The emergence of dCTA offers a promising solution through enhanced temporal resolution, allowing the visualization of real-time flow dynamics within the aneurysmal sac essential to establishing endoleak treatment or post-EVAR follow-up.

## 1. Introduction

The abdominal aortic aneurysm (AAA) is one of the major vascular diseases with a significant mortality risk associated with its rupture [1]. Endovascular aneurysm repair (EVAR) has revolutionized the treatment of AAA, offering a less invasive alternative to traditional open surgery [2]. This technique has gained popularity due to its lower perioperative mortality, faster recovery, and fewer complications, making it suitable for elderly patients or those with comorbidities [3,4,5]. Patients undergoing EVAR have a higher risk of reintervention than patients treated with open surgery [6], primarily for the management of complications such as endoleaks, defined as persistent blood flow within the aneurysm sac after EVAR [7]. For complex aortic anatomy chimney EVAR (CHEVAR), fenestrated or branched EVAR are being used more frequently [8]. Imaging-based surveillance after EVAR is therefore essential for the early detection, characterization, and localization of endoleaks to guide therapeutic intervention. It is crucial to identify endoleaks early to prevent the risk of rupture [9]. In most cases, endovascular techniques are the primary treatment approach for managing endoleaks [10,11]. Conventional imaging modalities, such as duplex ultrasound (DUS), contrast-enhanced computed tomography angiography (CTA), and magnetic resonance angiography (MRA), exhibit limitations in dynamic flow assessment, type differentiation, and sensitivity to low-flow endoleaks [12,13]. CTA with three phases (unenhanced, arterial, and delayed) is the main modality of surveillance with excellent sensitivity for detecting endoleaks [14]. However, this method may miss low-flow leaks, especially during the late arterial phase [15]. On the other hand, dynamic computed tomography angiography CT angiography (dCTA), also defined as four-dimensional computed tomography (4D-CT), closely follows the contrast bolus at multiple timepoints and may allow better characterization and localization of difficult-to-diagnose endoleaks. Dynamic CTA seems to be associated with the precise characterization of expanding aneurysmal sacs [16] and may offer a promising solution for the visualization of real-time flow dynamics within the aneurysmal sac. Moreover, in cases of complex fenestrated and/or branched repairs, multiple potential sources of endoleaks may exist and dCTA could identify the origin of the endoleak with the potential of simplifying the treatment [17]. This study aimed to consolidate the existing literature on the use of dCTA in detecting endoleaks, as well as its role in EVAR follow-up and its possible application for endoleak management. Finally, the goal is to offer a comprehensive review of the current applications of dCTA for endoleak detection and characterization after EVAR, aiding vascular specialists in selecting the most appropriate imaging modality for EVAR follow-up.

## 2. Materials and Methods

### 2.1. Data Sources, Search Strategy and Selection Criteria

This study was conducted in accordance with the Preferred Reporting Items for Systematic Reviews and Meta-Analyses (PRISMA) statement [18]. Papers on dCTA for the detection of endoleaks in patients who underwent endovascular repair of abdominal aortic aneurysms were searched in PubMed. We performed the last search on 1 November 2024, with no restrictions on the initial dates of the included studies. The following words were searched in PubMed: (“endoleak” or “endoleaks” or “evar”) and (“four-dimensional computed tomography” or “4d ct” or “4d computed tomography” or “dynamic cta” or “dynamic ct” or “dynamic computed tomography”). We included papers that reported the outcome of dCTA applications, excluding case reports or very limited case series (≤4). Articles in which dCTA was used to characterize different aortic neck dilatations or wall thickness changes were excluded. Two investigators (A.C. and C.B.M.) screened all titles and retrieved abstracts through the search strategy for relevance. The full texts of all relevant articles were obtained and reviewed for suitability independently by both reviewers (A.C., C.B.M.), and any disagreement in study inclusion was resolved by consensus and was also resolved by the senior author (P.P.) The studies included were full English texts. The references of all the included articles were reviewed to identify additional relevant studies for the comprehensive review. All studies reporting the (i) diagnostic imaging follow-up post-EVAR to rule out the presence of endoleaks, (ii) technical methodologies used to detect endoleaks, and (iii) performance of dynamic CTA in characterizing endoleaks in advanced endovascular aortic repair were included.

### 2.2. Data Extraction and Outcome

The extracted data included the first author, year of publication, study type, number of analyzed examinations, indications for endovascular procedures, type of procedure [endovascular aortic repair (EVAR), chimney endovascular aortic repair (CHEVAR), thoracic endovascular aneurysm repair (TEVAR), and fenestrated or branched endovascular aortic repair (FB-EVAR)], methodology used to detect endoleaks (dCTA), types of detected endoleaks, radiation, and/or contrast media dose for dCTA examinations. The data that could not be inferred were labeled “not extractable” (NE) or “not reported” (NR), as appropriate. We focused on the applications of dCTA in vascular surgery. Therefore, the main aspects and outcomes considered were as follows:-Endoleak detection rates-Characterization of endoleaks-Technological advantages-Radiation dose-Contrast agent used

### 2.3. Definitions

Endoleaks were defined according to the most recent European Society for Vascular Surgery (ESVS) Clinical Practice Guidelines and also from the Italian Society of Vascular and Endovascular Surgery (SICVE) as the presence of flow in the aneurysm sac outside the stent graft after EVAR and were classified as primary (present at the time of repair) or secondary (occurring after prior negative post operative imaging), as well as on the cause of peri-graft flow in five types [19,20,21]. The classification of endoleaks presented in our comprehensive review integrates the standard system outlined in the ESVS Clinical Practice Guidelines together with an alternative classification proposed by Oderich at.al, which is commonly employed in the literature for complex endografts [22]. The standard classification distinguishes endoleaks into five types: Type I (proximal or distal attachment site failure), Type II (retrograde flow into the aneurysm sac via branch vessels), Type III (fabric or modular disconnection failure), Type IV (graft porosity), and Type V (endotension) [19,23]. We also included a classification that refines the approach for complex endografts, defined as junctional failure in the modular components of branched or fenestrated devices [22]. We included the analysis of gutters after alternative treatment for complex thoraco-abdominal aneurysms, including CHEVAR and F-BEVAR. The gutters are due to a geometric mismatch between the endograft and additional side branch grafts and are a site of potential early type Ia endoleaks [8]. By combine the above classification, we aim to achieve a more comprehensive understanding of endoleak behavior and outcomes, thereby enhancing both diagnostic precision and therapeutic decision-making. We integrated the above-mentioned classifications, specifying the different causes of endoleaks after TEVAR [24] and complex EVAR (Table 1).

## 3. Results

Fourteen full texts of all relevant articles were obtained and reviewed. This study identified 33 papers. Of these, 12 were excluded based on title and abstract. Considering the remaining 21 papers, 20 were retrievable and were therefore reviewed in full-text form. After the exclusion of reviews (*n* = 1), case reports (*n* = 2), or small case series (3), 14 studies on the role of dCTA in endoleak detection and classification after aortic endovascular procedures were finally included in the comprehensive review (Figure 1). Detailed data regarding these results are reported in Table 2. All the analyzed studies were retrospective. Overall, 377 computed tomography angiographies were performed in 14 studies [8,12,17,25,26,27,28,29,30,31,32,33,34,35]. In three studies [17,25,26,27,28,29,30,31,32,33,34,35], aortic aneurysms treated with endovascular exclusion were thoraco-abdominal; these cases were managed using complex endografts, including fenestrated (FEVAR) and branched (BEVAR) endografts [17,25,26,27,28,29,30,31,32,33,34,35]. In one case, a juxtarenal aortic aneurysm was treated with chimney endovascular aneurysm repair (CHEVAR) [8]. In the studies considered, the first diagnostic examination following EVAR was standard CTA, followed by dynamic CTA to complete the diagnostic assessment. In only two studies, it was not explicitly reported whether CTA was employed as the initial diagnostic modality during follow-up after EVAR [8,26]. An example of the utility of dCTA in the research and analysis of endoleaks following endovascular treatment with the implantation of a thoraco-abdominal stent-graft (TEVAR and BEVAR) in a symptomatic Type B aortic dissection is shown in Figure 2.

### 3.1. Endoleak Detection and Characterization of Endoleaks

Overall, standard CTA detected 134 endoleaks [8,12,17,25,26,27,28,29,30,31,32,33,34,35]. In two cases the type of standard image acquisition was not reported [8,26]. In four studies, the specific types of endoleaks identified using standard CTA were not reported [8,26,30,32]. Among the reported cases, 95 endoleaks were classified as type II [12,17,25,27,28,29,31,33,35]. Type II endoleaks were suspected in two additional cases [30,32]. Type I endoleaks were detected in 20 cases [25,28,33,34,35], and type III endoleaks were detected in 17 cases [25,28,33,34]. Dynamic CTA was consistently selected as the diagnostic imaging technique for further assessment. Overall, with dCTA, 245 endoleaks were identified. Of these, 138 were type II endoleaks [12,17,25,27,28,29,30,31,32,33,35], 17 were type III [25,28,29,33,34], and 36 were type I [17,25,26,28,29,34,35]. One case reported an unspecified endoleak type [35], whereas another described an uncertain classification [17]. One study reported 44 endoleaks but did not specify the definition by type [31]. In particular, dCTA detected 36 type I endoleaks, compared with 16 identified with standard CTA; 138 type II endoleaks were found with dCTA, versus 95 with standard CTA; and 17 type III endoleaks and 12 gutters were detected with dCTA, compared with 17 with standard CTA. In Overeem et al.’s study, dCTA noted four gutters for each different endograft used. The gutters are a dynamic phenomenon, with a volume that changes throughout the cardiac cycles [8]. In particular, in Tarulli et al., in five cases of complex endograft (F-BEVAR), a type II and/or type III endoleak was suspected with standard CTA, but dCTA detected and confirmed type III endoleaks; in only one case, a type III endoleak was suspected, but it was revealed to be a type II endoleak on dynamic CT [17]. In the only study reporting the analysis of endoleaks after TEVAR, dCTA identified 16 endoleaks, compared to 7 detected with standard CTA. Among these, dCTA revealed two additional type Ia endoleaks, one type Ib, and four type IIIa endoleaks [35]. The same study reported that standard CTA identified fewer type II endoleaks after EVAR compared to dCTA (sCTA: *n* = 3 vs. dCTA: *n* = 19; *p* = 0.002) [35].

Dynamic CTA might also be useful as a guide for the treatment of endoleaks, improving the identification of target vessels in type II endoleaks and supported precise image-guided embolization. In Berczeli et al., nine patients underwent d-CTA-guided embolization with a median of 1 angiogram (range: 1–4) before the procedure [25].

### 3.2. Scanner Models Used for Image Acquisition

The studies employed a variety of CT scanner models from different manufacturers, including the following: Somatom Force (syngo.via®, VB30, Siemens Healthineers, Erlangen, Germany) was utilized in multiple studies [25,27,28,29]; Somatom Definition Flash (128-row CT scanner, Siemens Healthcare, Forchheim, Germany)) was utilized in two cases [26,31]; Siemens Leonardo (3-dimensional workstation, Siemens Medical Solutions, Erlangen, Germany) was used in one study [34]; Revolution GSI (AW Server 3.2; GE Medical System, Chicago, Illinois) was used in one study [12]; Aquilion ONE (320-row CT scanner, Canon Medical Systems, Tochigi, Japan) was used in two papers [32,35]; Brilliance iCT (256-slice CT scanner, Philips Healthcare, Eindhoven, Netherlands) was reported in one [8], in particular for dCTA; and Philips Medical Systems was also reported in one case [33].

### 3.3. Radiation Dose Analysis in CTA and dCTA Protocols

The radiation dose in CTA was not extractable in one article [17] and was not reported in eight papers [8,12,26,30,31,32,33]. The radiation dose in dCTA was not reported in three studies [12,27,29] and was not extractable in two articles [30,33]. In the studies that reported the radiation dose for CTA, the values ranged from 829 mGy/cm [27] to 1612.3 ± 530.3 mGy/cm [25]. Similarly, for dCTA, the dose ranged from 855.7 ± 54.2 mGy/cm [35] to an exceptionally high value of 4724 mGy/cm [17], which reflects a combination of clinical, technical, and procedural factors tailored to specific patient or diagnostic needs.

Berczeli et al. reported similar radiation doses for d-CTA (1445 ± 550 mGy cm) and CTA (1612 ± 530 mGy cm, *p* = 0.255), due to lower kV (80, 70–97.5 kV) and a smaller scan range (23–33 cm), despite multiple scans with d-CTA [25].

### 3.4. Contrast Media Utilization: Volume and Composition in CTA and dCTA

Different types of iodinated and non-iodinated contrast agents were used across the studies, where iodixanol 320 mg/mL, iomeprol 400 mg/mL, iobitridol 350 mg/mL, iopromide 370 mg/mL, and iopidamol 300 mg/mL. The contrast volume used for CTA ranged from as low as 15–20 mL [25,27,28], but these data were reported in three articles only [12,25,27,28]. For dynamic dCTA, higher contrast volumes were often used, with values up to 70–90 mL in some cases [25,26,31] or higher as 120 mL [33]. Not all studies provided detailed data on the exact type and volume of contrast media used in the dCTA [28,30], while data regarding CTA were reported in only three papers [25,27,28].

## 4. Discussion

Endoleak remains a critical post-procedural complication following EVAR for AAA. The treatment of different types of endoleaks differs significantly, both in terms of technical approach and clinical outcomes. Furthermore, the characterization of endoleaks may be particularly challenging after complex aortic procedures such as FB-EVAR or CHEVAR, where multiple grafts, branches, or fenestrations represent additional potential sources of endoleaks.

Therefore, accurate identification and characterization of endoleaks are essential to guide subsequent planning and management and follow-up [36,37]. In this comprehensive review, we analyzed 14 studies comparing dCTA with standard CTA imaging employing a three-phase acquisition protocol.

Dynamic CTA identified a total of 245 endoleaks, compared to 134 detected by CTA, suggesting that dCTA has a higher sensitivity for detecting endoleaks, potentially due to its ability to capture the temporal dynamics of contrast flow. Notably, two cases were either unspecified [35] or described as uncertain [17], and an additional study reported 44 endoleaks without specifying their type [31]. Both imaging modalities identified all major endoleak types (I, II, III), though dCTA reported higher numbers across all categories: dCTA detected 36 type I endoleaks, compared to 16 identified with standard CTA; 138 type II endoleaks were found with dCTA, versus 95 with standard CTA; and 17 type III endoleaks, and 12 gutters were detected with dCTA, compared to 17 with standard CTA. Dynamic CTA demonstrated a slight advantage in clarifying uncertain or unclassified endoleaks. This suggests that dynamic imaging may be particularly beneficial for identifying low-flow or intermittent endoleaks that might be missed on static imaging. These findings align with the current literature, which consistently reports type II endoleaks as the most frequent [38]. Type II endoleaks, often attributed to retrograde flow from collateral arteries, are typically considered less critical than type I or III due to their lower risk of aneurysm rupture [36]. However, their high prevalence underscores the need for meticulous surveillance, as persistent type II endoleaks may eventually increase the sac diameter and require intervention [39]. The limitation of standard CTA in detecting slow flow type II endoleaks further supports the use of dCTA during imaging follow-up after EVAR, with the ability to identify slow flow endoleaks through changes in the blood volume parametric map.

Berczeli et al. compared dCTA with standardized triphasic CTA in diagnosing endoleak types after EVAR using digital subtraction angiography (DSA) as the reference standard, reporting that the number of vessels causing type II endoleaks identified by dCTA, CTA, and DSA were 23, 17, and 16, respectively. This demonstrated that for type II endoleaks, dCTA better identified target vessels and enabled safe, targeted embolization [24].

In the majority of the studies included, the assessment of endoleaks using dCTA primarily relied on the analysis of temporal Hounsfield unit (HU) changes within the time-attenuation curves by drawing a region of interest [12,27,28,34]. This approach enables a precise evaluation of contrast dynamics and enhances the ability to characterize endoleak behavior through quantitative measurements. These measurements can be valuable in differentiating endoleak types and in providing an objective approach to endoleak diagnosis.

It is well-established in the literature that type I and III endoleaks can sometimes be occult, leading to the misdiagnosis of type II endoleaks that do not respond to treatment [40]. In some cases, type II endoleaks, which generally have a benign natural course, can be associated with sac enlargement. However, when there is rapid aortic sac growth (5 mm/year or more), or when attempts at treating type II endoleaks fail, the suspicion of delayed or occult type I or III endoleaks should be raised [41].

Rydberg at al. reported five patients in whom the CTA surveillance protocol could not differentiate between type II and type III endoleaks [33]. One of these cases involved a type III endoleak from a fabric tear that occurred where the lower part of the stent, near the top of the aneurysm sac, was pushed into the upper part of the stent due to a large, calcified plaque in the back of the aneurysm [33]. This highlights the importance of the correct analysis of computed tomography images from dCTA, combined with an accurate calcification measurement method, which could play an important role in decision-making and facilitate precise surgical planning by providing information on plaque morphology and its impact on the vessel lumen [42].

Failure of type II endoleak treatment in the presence of aneurysm sac growth or delayed rupture has been attributed to incomplete embolization of target vessels or an occult type I or III endoleak that was not initially diagnosed by imaging [43].

Dynamic CTA could be a useful adjunctive imaging modality for challenging or recurrent endoleaks in patients with complex endovascular aortic repairs [44]. For example, Tarulli et al. described how conventional CTA identified an endoleak of unclear origin near a fenestrated abdominal aortic endograft. Using dCTA, the flow and progression of contrast were clearly delineated, revealing a Type IIIc endoleak directly inferior to the left renal fenestration. With dCTA, it is possible to closely follow the contrast bolus at multiple timepoints, allowing better characterization and localization. This precise characterization facilitated accurate localization, aiding in targeted management [17].

After EVAR, one of the most clinically relevant procedure-related complications is stent-graft migration that can lead to repressurization of the aneurysmal sac, type I endoleak, aneurysm growth, and even rupture. The analysis of the forces leading to migration through dCTA is crucial for its prevention, diagnosis, and treatment [45].

Therefore, distal oversizing of up to 20% should be considered to reduce the risk of type I endoleak [46].

In this clinical context, the dCTA’s capability to capture continuous vascular dynamics across multiple phases demonstrated superior sensitivity in detecting the type of endoleaks, particularly in cases with unidentified or delayed filling and could offer enhanced visualization of feeding and draining vessels, potentially improving diagnostic precision, which is crucial for intervention or close surveillance.

Dynamic CTA was processed using scanners capable of rapid volumetric acquisitions with detailed image quality. The studies employed a variety of CT scanner models from different manufacturers, including: Siemens Healthineers (Somatom Force, Somatom Definition Flash, Siemens Leonardo), Canon Medical Systems (Aquilion ONE), GE Healthcare (Revolution GSI), and Philips Medical Systems (Brilliance iCT). The predominant use of Somatom Force and Somatom Definition Flash highlights a preference for high-end scanners offering speed and precision in diagnostic imaging [47]. Devices such as the Somatom Force and Aquilion ONE, specifically designed for dynamic imaging, facilitated higher sensitivity in detecting endoleaks compared to standard CTA and were preferred for patients with complex aneurysms as well [25,35,48]. This underlines the importance of technical capabilities in identifying low flow or intermittent endoleaks [49]. These devices deliver exceptional image clarity while minimizing artefacts, making them ideal for detecting intricate vascular details [50,51]. Advanced algorithms enhance image quality by reducing noise without compromising detail, enabling precise diagnostics even at low doses [47,52].

Features such as CARE Contrast optimize the use of contrast agents, ensuring better visualization of vascular structures while mitigating risks for patients with sensitivity to contrast media [47]. In the analyzed articles, different scanners with varying capabilities and configurations were employed, which could impact image quality, radiation dose, and the ability to acquire dynamic images. For instance, while the Somatom Force (Siemens Healthineers) is optimized to reduce radiation dose while maintaining high-quality imaging, other systems like the Brilliance iCT (Philips Healthcare) and Revolution GSI (GE Healthcare) may have specific protocols that do not achieve the same level of efficiency [53].

The adoption of newer technologies appears to enhance not only diagnostic sensitivity but also dose management and patient comfort [54,55]. However, the variability in protocols and systems used across studies highlights the lack of methodological uniformity while pointing to a clear trend toward utilizing advanced technology for dCTA imaging.

Some authors emphasized the incremental radiation dose compared to standard CTA if diagnostic yield was deemed clinically justifiable [56]. However, advancements in acquisition protocols and software optimization could reduce these concerns, achieving a balance between diagnostic benefits and patient safety [55]. Dynamic CTA generally required higher radiation doses than standard CTA, as seen in most studies that reported both values [25,27]. However, some studies reported lower doses for dCTA compared to CTA [28], possibly reflecting protocol adjustments or scanner optimizations. Extended protocols with multiple phases or prolonged acquisition times to enhance temporal resolution might contribute to the higher dose. Specific diagnostic objectives like type III endoleak require finer spatial and temporal resolution and might demand higher radiation exposure. In fact, the highest ionizing radiation dose was observed in the article where the endovascular treatment of the aneurysm was more complex with F-BEVAR [17].

Some studies did not provide specific data on radiation dose, complicating uniform comparisons.

Different types of iodinated contrast agents were used across the studies. This indicates heterogeneity in the choice of contrast agents, possibly reflecting institutional preferences or availability. For many studies, the same contrast agent and similar volumes were used for both standard CTA and dCTA, reflecting consistency in imaging protocols within studies [27]. Not all studies provided detailed data on the exact type and volume of contrast media used [25,30], which limits direct comparisons. This underscores the need for standardization not only of the radiation dose but also of the quantity and quality of the contrast agent, which could be included in specific protocols for performing dCTA. The variability in contrast volume might impact diagnostic efficacy, especially for dCTA, where higher volumes may enhance the detection of understated endoleaks. The use of low volumes in certain studies could reflect efforts to minimize contrast-related complications in patients with compromised renal function [25,57].

In the literature, alternative methods for identifying endoleaks are well documented. Among these, angiography is a valuable diagnostic tool as it also enables immediate treatment of endoleaks, such as embolization in type II endoleaks. However, angiography is more invasive and carries higher risks for the patient. Among less invasive diagnostic investigations, contrast-enhanced ultrasound (CEUS) is an option, but it requires specific expertise in image interpretation. After conventional EVAR, the use of CEUS may offer some advantages over dCTA, in particular, the absence of radiation and the use of a non-nephrotoxic contrast agent. However, CEUS usage is limited during TEVAR or FB-EVAR follow-up [58].

In patients with a hostile abdomen, it may not be conclusive, as it does not visualize the structure of the endograft or assess the sealing zones. Nevertheless, this procedure is feasible and could detect more endoleaks compared to dynamic CTA, particularly in type II endoleaks [59,60].

A limitation of our comprehensive review is that the studies do not provide data on the comparative sensitivity or specificity of dCTA versus ultrasound or other imaging modalities, preventing a statistical comparison. However, as this is a relatively new imaging modality, such studies may become available in the future.

This study has other limitations. It is only a comprehensive review and the included papers present highly heterogeneous data regarding the main aspects and outcomes analyzed. There are only a few articles in the literature reporting case studies with large sample sizes, making it very challenging to standardize the results. Furthermore, the studies utilized technologies and scanners that differed significantly from one another. Dynamic TCA is a new and not widely adopted technique; consequently, the protocols used in the centers where this diagnostic investigation is available vary significantly and are difficult to compare. We created only a database summarizing the aspects and data most frequently presented and discussed in the available literature.

The ability to correlate endoleak presence with aneurysm sac expansion helps to stratify patients into those requiring urgent reintervention or continued surveillance. Dynamic CTA findings often influence decision-making, such as identifying endoleaks requiring embolization or reintervention, especially in challenging configurations involving fenestrated or branched endografts [61]. Incorporating dynamic CT into follow-up could reduce the likelihood of missed or underestimated endoleaks, potentially minimizing the risk of late rupture. However, this must be balanced with considerations of radiation exposure and contrast-induced nephropathy [62,63]. This diagnostic investigation could play an important role in cases of uncertain endoleaks before treatment or when the aneurysmal sac continues to expand even after treatment. Further research in this area promises not only improving the detection of endoleaks, which represent one of the most significant complications both in the early and late period following EVAR, but also improving the development of standardized protocols for the use of dCTA in the clinical practice. Enhanced detection capabilities could lead to earlier interventions and better long-term outcomes for patients.

## 5. Conclusions

In conclusion, dCTA emerges as a promising modality for the comprehensive evaluation of endoleaks following EVAR, particularly for complex cases requiring detailed anatomical and hemodynamic insights. The enhanced detection capability of dCTA, particularly for type II and low-flow endoleaks, underscores its role as a valuable follow-up imaging modality for patients undergoing EVAR. Its ability to identify additional endoleaks may influence patient management, such as the need for further intervention or close surveillance. The lack of standardized reporting in retrospective analyses hinders comparisons and emphasizes the need for uniform protocols. Future research should focus on refining protocols and validating cost-effectiveness to establish this modality as a routine diagnostic tool.

## Figures and Tables

**Figure 1 diagnostics-15-00370-f001:**
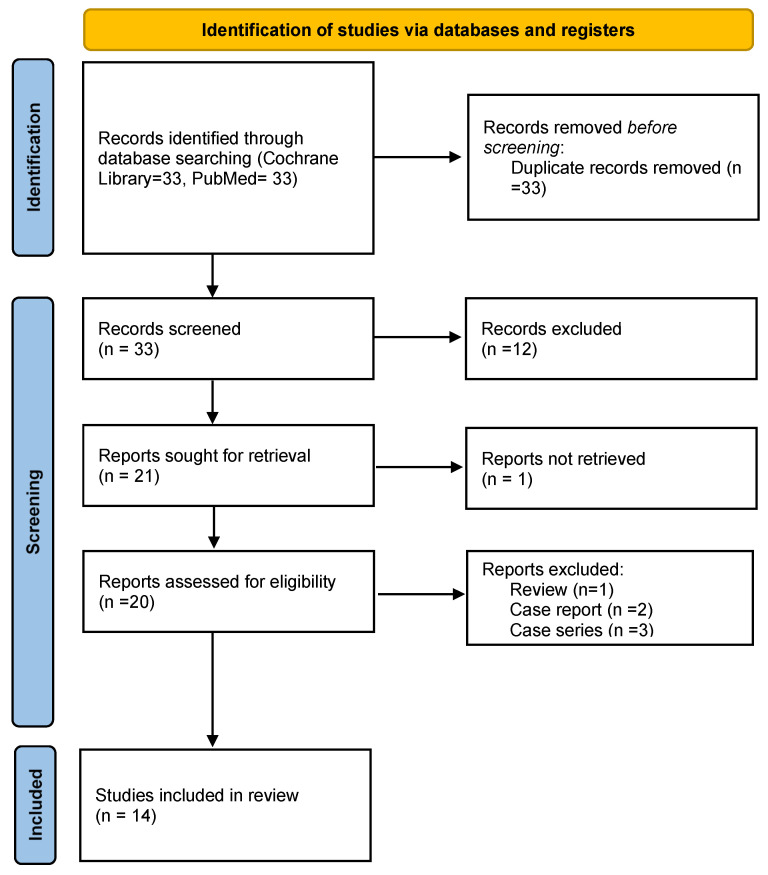
PRISMA flow diagram showing the identification process of the included studies. PRISMA, Preferred Reporting Items for Systematic Reviews and Meta-Analysis.

**Figure 2 diagnostics-15-00370-f002:**
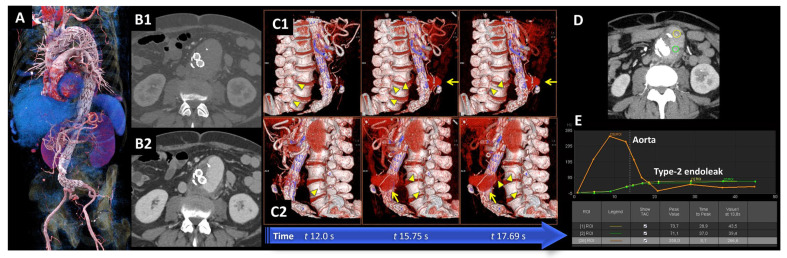
(**A**). 3D CT angiography (CTA) image of a 57-year-old man who underwent three-stage thoraco-abdominal stent-graft aortic repair (TEVAR and BEVAR) for symptomatic Type B aortic dissection. (**B**). At 3 month-follow-up, a CTA was performed for abdominal pain in the emergency setting demonstrating the presence of a large endoleak with a significant increase in sac size from 53 mm to 71 mm in maximum anterior–posterior diameter. However, the standard static CTA, including arterial and venous acquisitions ((**B1**) and (**B2**), respectively), was insufficient to further characterize the endoleak type. (**C**). Dynamic or 4D CTA (4D-CTA) was performed using a second-generation CT scanner (Somatom Definition Flash^®^, Siemens Healthineers, Forchheim, Germany), which acquired multiple time-resolved contrast enhanced scans (11 scans in this specific case) during table movement in shuttle mode. 4D-CTA three-dimensional reconstructions revealed a large type-2 endoleak (arrows) from the third lumbar arteries (arrowheads in (**C1**) and (**C2**)). Note that the contrast opacification occurs earlier and is more pronounced in the right lumbar artery (arrowheads in (**C1**)) compared to the left lumbar artery (arrowheads in (**C2**)). Additionally, observe the large type II endoleak extending to the anterior aspect of the aneurysmal sac (arrows in (**C1**) and (**C2**)). (**D**,**E**). 4D-CTA image dataset was evaluated qualitatively and quantitatively after motion correction and 3D noise reduction using dedicated software VB60A_HF06 (syngo.via^®^, Siemens Healthineers, Forchheim, Germany) that generates time-attenuation curves (TACs) for the analysis of the resulting temporal changes in contrast enhancement within regions of interest (**D**). In (**E**), note the different phenotype of the aortic TAC (orange curve) compared to the endoleak TACs (yellow and green curves), the latter presenting a gradual upslope, a wide plateau, a low peak value, and gradual endoleak washout due to slow outflow. TAC analysis, incorporating the peak value and time-to-peak parameters, enhances the characterization of endoleaks type and their inflow and outflow patterns. This approach overcomes the limitations of the standard thriphasic (non-contrast, arterial, and delayed phases) CTA improving the sensitivity and specificity for the detection and characterization of endoleaks, with a comparable or lower radiation dose profile (dose length product [DLP] of the static thriphasic CTA acquisition of 1329 mGy*cm and of the 11-phase 4D-CTA of 835 mGy*cm).

**Table 1 diagnostics-15-00370-t001:** Comprehensive classification of endoleaks after EVAR.

Endoleak Type	Subtype	Description
Type I	Ia	Perigraft flow due to an incomplete seal at the proximal attachment site
	Ib	Perigraft flow due to an incomplete seal at the distal attachment site
Type I for F-BEVAR or TEVAR	Ic	Perigraft flow due to an incomplete seal at the sidebranch attachment
Type II		Retrograde flow from collateral vessels, such as the lumbar or inferior mesenteric arteries
Type III		Graft material failure or modular disconnection, resulting in leakage through defects in the prosthesis/junctional failure in modular components
Type III for F-BEVAR or TEVAR	IIIa	Junctional failure in aortic–aortic or aortic–bifurcated or bifurcated–iliac modular components
	IIIb	Junctional failure in fabric tear or fracture
	IIIc	Junctional failure in attachment aortic side branch–side branch component
Type IV		Porosity of the graft material, typically transient and observed early postoperatively
Type V		Characterized by sac expansion without radiologically visible endoleaks (also known as endotension)

F-BEVAR, fenestrated-branched endovascular aortic repair; TEVAR, endovascular thoracic aortic repair.

**Table 2 diagnostics-15-00370-t002:** Detailed data about results.

Author	Year	Type	Number Analyzed AngioTC	Aneurysm Type	Procedure	Standard Image Acquisition	EL Detected (*n*)	EL Type (*n*)	Different Image Acquisition	EL Detected (*n*)	EL Type (*n*)	Software Used	Radiation Dose with CTA mGy/cm	Radiation Dose with dCTAmGy/cm	Total Contrast Used with CTA	Total Contrast Used with dCTA
Apfaltrer [27]	2020	R	19	AAA	EVAR	CTA	9	II	dCTA	11	II	Somatom Force, Syngo.via® VB10B, Siemens Healthineers, Forchheim, Germany	829 (596.8–994.3)	1063.8 (1063.6-1064.9)	50 mL Iohexol (350 mgI/mL)	50 mL of Iohexol (350 mgI/mL)
Berczeli [28]	2021	R	4	AAA	EVAR	CTA	5	II	dCTA	5	I–(3) II(1)-III(1)	Somatom Force syngo.via®, VB30, Siemens Healthineers, Erlangen, Germany	1123.8 ± 384.1	900.1± 126	20 mL Iodixanol (320 mg/mL)	NR
Berczeli [29]	2022	R	24	AAA	EVAR	CTA	23	I (4), II (16), III (2)	dCTA	23	I (4)–II(16)–III (3)	Somatom Force syngo.via®, VB30, Siemens Healthineers, Erlangen, Germany	1038 ± 533	NR	NR	77 mL iodixanol (320 mg/mL)
Berczeli [25]	2023	R	52	AAA, AATA	EVAR, FEVAR	CTA	19	I(4), II (14), III (1)	dTCA	19	I (4), I (14) e III (1)	Somatom Force syngo.via®, VB30, Siemens Healthineers, Erlangen, Germany	1612.3 ± 530.3	1445 ± 550.5	15–20 mL iodixanol (320 mg/mL)	70–90 mL iodinated (not specified)
Charalambous [12]	2020	R	9	AAA	EVAR	CTA	9	II	dTCA	9	II	Revolution GSI, AW Server 3.2; GE Medical System, Chicago, Illinois	NR	NR	NR	60 mL iopromide media (370 mg/mL)
Haubenreisser [30]	2015	R	54	AAA	EVAR	CTA	NR	suspected II	dTCA	19	II	NR	NR	NE	NR	NR
Lehmkuhl [31]	2013	R	72	AAA	EVAR	CTA	24	II	dTCA	44	I–II–III	Somatom Definition Flash, 128-row CT scanner, Siemens Healthcare, Forchheim, Germany)	NR	1344 ± 131	NR	80 mL iopremol (400 mg/mL)
Lehmkuhl [26]	2012	R	21	AAA	EVAR	NR	NR	NR	dTCA	26	I (1), II (25)	Somatom Definition Flash, Flash, 128-row CT scanner, Siemens, Forchheim, Germany	NR	1293 ± 104	NR	80 mL iopremol (400 mg/mL)
Nishihara [32]	2020	R	10	AAA	EVAR	CTA	NR	suspected II	dTCA	10	II	Aquilion ONE, 320-row CT scanner, Canon Medical Systems, Tochigi, Japan	NR	the same as standard CTA	NR	Iopamidol (not specified)
Overeem [8]	2018	R	3	AAA Juxtarenal	CHEVAR	NR	NR	NR	dTCA	12	Gutters	Brilliance iCT, 256-slice CT scanner, Philips Healthcare, Eindhoven, Netherlands		NR	NR	Iobiditrol (350 mg/mL), (not specified)
Rydberg [33]	2004	R	12	AAA	EVAR	CTA	12	I (1), II (10), III (1)	dTCA	5	II (4), III (1)	Mx8000IDT16, 16-channel scanner, Philips Medical Systems, Cleveland, Ohio, USA	NR	NE	NR	120 mL Iopidamol(300 mg/mL)
Tarulli [17]	2022	R	13	AATA	FB-EVAR	CTA	16	III (10), II (4), I (2)	dTCA	12	III (8), II (2), I(1), uncertain (I)	Aquilon One Prism, 320 slice scanner, Canon Medical Systems Corporation, Otawara, Tochigi, Japan	NE	4724	NR	20 mL Iodixanol or Iopromide (320 mg/mL or 300 mg/mL)
Waldeck [35]	2022	R	69	AATA	EVAR-TEVAR	CTA	11	I (8), II (4),	dTCA	44	I (20), II (23), no specification (1)	Aquilion ONE, 320-row CT scanner, Toshiba Medical Systems, Otawara, Japan)	NR	855.7 ± 54.2	NR	60–70 mL Iohexol (350 mg/mL)
Sommer [34]	2010	R	15	AAA	EVAR	CTA	6	III (3), I (3)	dTCA	6	III (3), I (3)	Siemens Leonardo, 3-dimensional workstation, Siemens Medical Solutions, Erlangen, Germany	1393 ± 290 (triphasic CTA), 953 ± 143 (biphasic CTA)	902 ± 63	NR	60 mL Iomeprol (400 mg/mL)

NR, not reported; NE not extractable; R, retrospective; AAA, abdominal aortic aneurysm; AATA, thoraco abdominal aneurysm, EVAR, endovascular aortic repair; TEVAR, thoracic endovascular aneurysm repair; CHEAVR, chimney endovascular aortic repair; FB-EVAR, Fenestrated or branched endovascular aortic repair; CTA, computed thomography angiography; dCTA, dynamic computed thomography angiography; n, number.

## Data Availability

The original contributions presented in the study can be directed to the corresponding author. No additional supplementary material or publicly archived datasets are available for this study due to privacy and ethical considerations.

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
