# Peer review of "The Role of Dynamic Computed Tomography Angiography in Endoleak Detection and Classification After Endovascular Aneurysm Repair: A Comprehensive Review"

_diagnostics, 2025, doi:10.3390/diagnostics15030370_

Round 1
Reviewer 1 Report
Comments and Suggestions for Authors
I congratulate with the Authors for their review about the role of dCTA in detecting EL after endovascular treatment of aortic pathologies.
For a better understanding of your research I would suggest to separate the results of conventional EVAR from those of TEVAR and of CH-EVAR/FEVAR/BEVAR (and therefore the respective classifications of EL), because they are different pathologies and there are different considerations to be done. For example after conventional EVAR the use of contrast-enhanced DUS may offer some advantages over dCTA.
Author Response
Comment:
I congratulate with the Authors for their review about the role of dCTA in detecting EL after endovascular treatment of aortic pathologies.
For a better understanding of your research I would suggest to separate the results of conventional EVAR from those of TEVAR and of CH-EVAR/FEVAR/BEVAR (and therefore the respective classifications of EL), because they are different pathologies and there are different considerations to be done. For example after conventional EVAR the use of contrast-enhanced DUS may offer some advantages over dCTA.
Response:
Thank you very much for pointing out the importance of paying attention to this significant aspect. We have updated our table on the classification of endoleaks by introducing the characteristics of endoleaks after TEVAR and complex EVAR. This aspect has been added at lines 148-150 and in the table 1. Only one study analyzed endoleaks after TEVAR using dCTA (Ref 35). We presented these findings separately from the global data on endoleaks after TEVAR, as recommended by the reviewer, this aspect has been added at lines 206-209: “in the only study reporting the analysis of endoleaks after TEVAR, dCTA identified 16 endoleaks, compared to 7 detected with standard CTA. Among these, dCTA revealed 2 additional type Ia endoleaks, 1 type Ib, and 4 type IIIa endoleaks”. We have also discussed the utility of DUS/CEUS after conventional EVAR compared to more complex intervention like F-BEVAR. This aspect has been added at line 437-440. Data about endoleak characterization after FEVAR and BEVAR are already reported in the text at lines 200etc. in the results, and discussed in lines 357 etc. The characterization of endoleaks becomes more challenging in FB-EVAR procedures due the growing complexity involving multiple grafts, branches, or fenestrations that create multiple potential sources of endoleak. We focused better at the beginning of the discussion (see lines 294-297) that FB-EVAR are different pathologies consequently the endoleak type are different and more difficult to detect. By using dCTA it is possible to closely follow the contrast bolus at multiple timepoints, allowing better characterization and localization less common endoleak like types III. We have explained more clearly this aspect and at line 360-361. Regarding the treatment with ChEVAR only one article one of the included articles analysed endoleaks after ChEVAR, thus data are insufficient to draw “strong” conclusions regarding this specific treatment (Ref. 8).
Reviewer 2 Report
Comments and Suggestions for Authors
The authors have done a great and important job.
I want to congratulate the authors on the excellent result.
dCTA acts as a very promising technology for a comprehensive assessment of endolic processing after EVAR. Its ability to identify additional endoleaks could change the clinical strategy for treating patients.
This is a very serious problem of endovascular surgery for aneurysms not only in the abdominal aorta, but also in other parts throughout the aorta. The occurrence of this complication can level all efforts for minimally invasive treatment of a severe cohort of patients with aortic aneurysms.
The authors devoted this work to this and discuss in it a promising new technology for the accurate diagnosis of complications of endovascular intervention. But this is a review! Formally, it cannot be either original or unoriginal. It's complete and logical.
The topic of Dynamic Computed Tomography Angiography is becoming more and more popular among surgeons, cardiologists and diagnosticians, and there are still many questions related to this technology! Any publication on this topic is currently relevant and interesting. Moreover, such work as peer-reviewed. It analyzes the positive aspects of this diagnostic method Endoleak Detection After EVAR and discusses unresolved and unclear issues that accompany Dynamic Computed Tomography Angiography at the modern level.
The conclusion corresponds to what is said in the work, and it is very correct.
The references are appropriate, accurate and modern.
Tables and figures illustrate the text of the article well and compactly supplement it.
Author Response
Comment:
The authors have done a great and important job.
I want to congratulate the authors on the excellent result.
dCTA acts as a very promising technology for a comprehensive assessment of endolic processing after EVAR. Its ability to identify additional endoleaks could change the clinical strategy for treating patients.
This is a very serious problem of endovascular surgery for aneurysms not only in the abdominal aorta, but also in other parts throughout the aorta. The occurrence of this complication can level all efforts for minimally invasive treatment of a severe cohort of patients with aortic aneurysms.
The authors devoted this work to this and discuss in it a promising new technology for the accurate diagnosis of complications of endovascular intervention. But this is a review! Formally, it cannot be either original or unoriginal. It's complete and logical.
The topic of Dynamic Computed Tomography Angiography is becoming more and more popular among surgeons, cardiologists and diagnosticians, and there are still many questions related to this technology! Any publication on this topic is currently relevant and interesting. Moreover, such work as peer-reviewed. It analyzes the positive aspects of this diagnostic method Endoleak Detection After EVAR and discusses unresolved and unclear issues that accompany Dynamic Computed Tomography Angiography at the modern level.
The conclusion corresponds to what is said in the work, and it is very correct.
The references are appropriate, accurate and modern.
Tables and figures illustrate the text of the article well and compactly supplement it.
Response:
Thank you very much for your thoughtful and encouraging comment on our work. We are truly honored by your words and your recognition of the potential role of dCTA to improve the detection of endoleak after EVAR. Your acknowledgment of our comprehensive review and its contribution to the growing discussion around dCTA is highly motivating. As you correctly observed, patients undergoing this treatment represent a complex population requiring meticulous follow-up, making it essential to stay continuously updated on the most effective diagnostic approaches. Once again, thank you for your valuable feedback and for recognizing the importance of this topic. We will continue to explore and discuss this promising technology and hope that our work contributes to further advancements in this field.
Reviewer 3 Report
Comments and Suggestions for Authors
A review of CTa to diagnose endoleaks
The results should mention how accurate this is (eg better or worse than ultrasound). Comparing the raw number is inadequate to compare the two - this should be a rate.
The introduction is quite lengthy and should be condensed for clarity.
Author Response
Comments 1 :
The results should mention how accurate this is (eg better or worse than ultrasound). Comparing the raw number is inadequate to compare the two - this should be a rate.
Response 1:
We appreciate your valuable feedback regarding the need to express diagnostic accuracy as rates rather than raw number. Unfortunately, comparative studies regarding the use of dCTA vs US (or other imaging modalities) are currently lacking in the literature. Moreover, synthesizing such data is beyond the scope of our comprehensive review. We acknowledge this as a limitation and have addressed it in the limitations paragraph. Please see lines 445-448. However, we anticipate that such studies will likely become available in the future.
In respose to the Reviewer’s comment, we have revised the discussion section as follows:
Lines 445-448: A limitation of our comprehensive review is that the included studies do not provide data on the comparative sensitivity or specificity of dCTA versus ultrasound or other imaging modalities, preventing a statistical comparison. However, as this is a relatively new imaging modality, such studies may become available in the future.
Comments 2:
The introduction is quite lengthy and should be condensed for clarity.
Response 2:
Thank you for your suggestion. We have shortened the introduction to improve clarity and focus. A total of 84 words were deleted to streamline the section.
Round 2
Reviewer 1 Report
Comments and Suggestions for Authors
The Authors have addressed all main issues.
Author Response
Comment:
The Authors have addressed all main issues.
Response:
We appreciate the reviewer’s thorough evaluation and are pleased that the main issues have been addressed to their satisfaction. Thank you for your valuable feedback, which has helped improve the quality of our manuscript.
Reviewer 3 Report
Comments and Suggestions for Authors
Improved compared to the original manuscript, but the results section still could use some elaboration.
Comments on the Quality of English Languagethe English language needs minor edits for readability
Author Response
Comment:
Improved compared to the original manuscript, but the results section still could use some elaboration.
The English language needs minor edits for readability
Response:
Thank you for pointing out the need for further elaboration and minor English edits to improve the readability of the Results section. We completely revised the text, and we checked and corrected the minor English issues (please see the changes in the track changes version of the manuscript). We also expanded the results section, as suggested by the Reviewer.
Please see:
Redline L214-220: The same study reported that standard CTA identified fewer type II endoleaks after EVAR compared to dCTA (sCTA: n=3 vs dCTA: n=19; p=0.002). [35] Dynamic CTA might also useful as a guide for the treatment of endoleaks im-proving the identification of target vessels in type II endoleaks and supported precise image-guided embolization. In Berczeli et al., nine patients underwent d-CTA-guided embolization with a median of 1 angiogram (range: 1-4) before the procedure. [24]
Redline L238-240: Berczeli et al. reported similar radiation doses for d-CTA (1,445 ± 550 mGy cm) and CTA (1,612 ± 530 mGy cm, P = 0.255), thanks to lower kV (80, 70-97.5 kV) and a smaller scan range (23-33 cm), despite multiple scans with d-CTA. [24]